# E6/E7 Variants of Human Papillomavirus 16 Associated with Cervical Carcinoma in Women in Southern Mexico

**DOI:** 10.3390/pathogens10060773

**Published:** 2021-06-20

**Authors:** Ramón Antaño-Arias, Oscar Del Moral-Hernández, Julio Ortiz-Ortiz, Luz Del Carmen Alarcón-Romero, Jorge Adán Navor-Hernández, Marco Antonio Leyva-Vázquez, Marco Antonio Jiménez-López, Jorge Organista-Nava, Berenice Illades-Aguiar

**Affiliations:** 1Laboratorio de Biomedicina Molecular de la Facultad de Ciencias Químico Biológicas, Universidad Autónoma de Guerrero, Chilpancingo, Guerrero CP 39090, Mexico; ramon.a@uagro.mx (R.A.-A.); julioortiz@uagro.mx (J.O.-O.); anavor@uagro.mx (J.A.N.-H.); 02313@uagro.mx (M.A.L.-V.); jorgeorganista@uagro.mx (J.O.-N.); 2Laboratorio de Virología de la Facultad de Ciencias Químico Biológicas, Universidad Autónoma de Guerrero, Chilpancingo, Guerrero CP 39090, Mexico; odelmoral@uagro.mx; 3Laboratorio de Citopatología e Histoquímica de la Facultad de Ciencias Químico Biológicas, Universidad Autónoma de Guerrero, Chilpancingo, Guerrero CP 39090, Mexico; lcalarcon@uagro.mx; 4Instituto Estatal de Cancerología “Dr. Arturo Beltrán Ortega” Acapulco, Guerrero CP 39570, Mexico; director@cancerologiagro.gob.mx

**Keywords:** human papillomavirus 16, E6, E7, cervical cancer, squamous intraepithelial lesion, E7 non-synonymous mutation

## Abstract

Persistent infection with the human papillomavirus 16 (HPV 16) is the cause of half of all cervical carcinomas (CC) cases. Moreover, mutations in the oncoproteins E6 and E7 are associated with CC development. In this study, E6/E7 variants circulating in southern Mexico and their association with CC and its precursor lesions were evaluated. In total, 190 DNA samples were obtained from scrapes and cervical biopsies of women with HPV 16 out of which 61 are from patients with CC, 6 from patients with high-grade squamous intraepithelial lesions (HSIL), 68 from patients with low-grade squamous intraepithelial lesions (LSIL), and 55 from patients without intraepithelial lesions. For all E7 variants found, the E7-C732/C789/G795 variant (with three silent mutations) was associated with the highest risk of CC (odd ratio (OR) = 3.79, 95% confidence interval (CI) = 1.46–9.85). The analysis of E6/E7 bicistron conferred to AA-a*E7-C732/C789/G795 variants revealed the greatest increased risk of CC (OR = 110, 95% CI = 6.04–2001.3), followed by AA-c*E7-C732/C789/G795 and A176/G350*E7-p. These results highlight the importance of analyzing the combinations of E6/E7 variants in HPV 16 infection and suggest that AA-a*E7-C732/C789/G795, AA-c*E7-C732/C789/G795, and A176/G350*E7-p can be useful markers for predicting CC development.

## 1. Introduction

Persistent infection with high-risk human papillomavirus (HR-HPV) is the main cause of cervical carcinoma (CC) and its precursor lesions [1]. Among the HR-HPV types, HPV 16 causes more than half of CC cases worldwide [2,3].

The oncogenicity of HPV 16 is because of the oncoproteins E6 and E7, which promotes cell transformation and immortalization; increase cell proliferation; invasion and migration; and supports immune evasion. E6 induces p53 degradation and prevents cell growth arrest and apoptosis. E7 binds to retinoblastoma proteins p107 and p130 and causes immortalization and deregulation of cell proliferation and differentiation. Hence, these oncoproteins act together to promote cervical carcinogenesis [4,5,6].

E6 and E7 genes are located in tandem at the beginning of the HPV 16 genome at nucleotides 104–559 and 562–858, respectively [7], and are initially transcribed as a single bicistronic HPV 16 E6/E7 pre-mRNA. Mature mRNA from the unspliced HPV 16 E6/E7 bicistron encodes both oncoproteins, spliced transcripts, and E6*I/E7 and E6*II/E7, which are short forms of E6 with intact E7 [8,9].

HPV 16 belongs to the genus Alphapapillomavirus, species group A9 [10]. It has intratypic variants associated with premalignant lesion progression and CC development [11,12,13,14]. HPV 16 variants have been classified into four lineages: A (European-Asian, EAS), B (African 1, AF1), C (African 2, AF2), and D (North-American/Asian-American, NA/AA). Furthermore, nine sublineages have been described: A1, A2, A3 (European, E), A4 (Asian, As), B1 (Afr1a), B2 (Afr1b), D1 (North-American, NA), D2 (Asian-American, AA1), and D3 (Asian-American, AA2) [12,15]. Most infections worldwide are caused by sublineages A1, A2, and A3 [15,16]. However, sublineage D2 is closely associated with the risk of CC [14].

In Mexico, several E6 variants of HPV 16 have been characterized. In Mexico City, the variants E-G131 (A1) E-G350, E-C188, E-A176 (A2), NA1 (D1), AA-a, and AA-c (D2) were reported and among them the AA variants are more oncogenic than the E variants [17]. In other studies, only the E-G350 (A2) variant was reported [18,19]. In southeastern Mexico, E-Prototype (A1), E-G350 (A2), Af2 (C), and AA (D2) were reported in samples from women with CC [20]. Moreover, in southern Mexico, we previously detected 27 E6 variants of HPV 16 and among them the most frequent and most closely associated with the development of CC were E-G350, E-C188/G350, E-176/G350 (A2), and AA-a and AA-c (D2) [21]. These HPV 16 E6 variants alter the expression of genes involved in proliferation, differentiation, apoptosis, cell cycle, adhesion, angiogenesis, transcription, and protein translation in C33-A cells and are consistent with the oncogenic potential of HPV 16 variants [22].

To date, few studies have focused on the intratypical variants of E7 as compared with those on E6 variants. Nevertheless, several studies demonstrated that non-synonymous mutations in E7 gene can cause amino acid changes. In particular, the variants A647G (NS29) and C749T (S63F) are associated with the progression of premalignant lesions into CC [23,24,25]. However, in a large study, E7 conservation was found to be significantly associated with CC, whereas E7 mutations were associated with reduced carcinogenicity of HPV 16 [13].

Taken together, a deeper understanding of the regional variants of HPV 16 is needed owing to their important epidemiological, preventive, and therapeutic implications for CC. Hence, the present study had aimed to perform the following: (1) to characterize the E7 genetic variations (since E6 variants circulating in southern Mexico have already been reported); (2) to determine the combinations of E6 and E7 variants in the HPV 16 E6/E7 bicistron in women from southern Mexico; and (3) evaluate the associations between the observed variants alone and in combination with the development of CC and precursor lesions.

## 2. Results

A total of 190 HPV 16 positive women (aged between 17 and 97 years) were included in the study. All participants were residents from the seven regions of the Guerrero state in southern Mexico. The average age of women with CC, high-grade squamous intraepithelial lesions (HSIL), low-grade squamous intraepithelial lesions (LSIL), and without intraepithelial lesions was 50.5, 36.3, 30.2, and 35.9 years, respectively. The clinical and demographic characteristics of the patients are presented in Table 1.

### 2.1. E7 Variants of HPV 16

E7 variants in HPV 16 identified in women in southern Mexico with CC, HSIL, LSIL, and non-intraepithelial lesions were analyzed. Eight E7 mutations were found and comprises two missense mutations (A647G and C712A) and six synonymous mutations (G666A, T678C, T732C, C765T, T789C, and T795G). The E7 prototype was detected most frequently (74.74%), followed by the E7-C732/C789/G795 variant (20.53%) with three silent mutations and a genotype that was associated with the highest risk of CC (odd ratio (OR) = 3.79, 95% confidence interval (CI) = 1.46–9.85). Other variants with synonymous mutations were detected, including E7-A666 (2.11%), E7-C678 (0.53%), and E7-T765 (0.53%). Variants with non-synonymous mutations were found in 2.11% (A712) and 0.53% (G647) of the samples (Table 2).

### 2.2. E6/E7 HPV 16 Bicistronic Variants of HPV 16

E6 and E7 variant combinations in the HPV E6/E7 bicistron were analyzed and grouped by sublineage. Four of the nine HPV 16 sublineages were present in the studied population: A1 (12.63%), A2 (66.84%), C1 (0.53%), and D2 (20%).

Mutations in the E6 and E7 genes of HPV 16 were identified in the same patient, and 34 bicistron combinations were detected. The most frequent combination of E6/E7 variants in HPV 16 was the E-G350*E7-prototype (36.8%), which belongs to the A2 sublineage, and the second most frequent combination was AA-a*E7-C732/C789/G795 (10%), which belongs to the D2 sublineage. Only 9.5% of the samples had a combination of E6 and E7 prototypes (Table 3).

In order to determine the effect of the combination of E6 and E7 variants in women infected with HPV 16, the risk of CC and its precursor lesions was analyzed. The risk of CC was highest in individuals harboring AA-a*E7-C732/C789/G795 (OR = 110, 95% CI = 6.04–2001.3), AA-c*E7-C732/C789/G795 (OR = 35, 95% CI = 2.63–465.37), and E-A176/G350*E7-prototype (OR = 23.33, 95% CI = 1.99–273.3). The oncogenic risk of the most common bicistronic variant, the E-G350*E7-prototype, was not statistically significant. The risk of developing CC precursor lesions was not significant for any of the E6/E7 combinations. The E-prototype*E7-prototype bicistron was used as a reference (Table 3).

## 3. Discussion

The aim of this study was to improve our understanding of HPV 16 variants circulating in southern Mexico owing to its epidemiological, preventive, and therapeutic implications for CC. We have previously reported that the most frequent E6 variants of HPV 16 and those associated with a high risk of developing CC were the European A1 and A2 sublineages (E-G350, E-prototype, E-A176/G350, and E-C188/G350) and Asian-American D2 (AA-a and AA-c) lineages [21]. This is the first study of E7 variants in HPV 16 in Mexico involving the analyses of variant combinations in the HPV E6/E7 bicistron in cervical samples as well as their association with the development of CC and its precursor lesions.

Herein, fewer variants were detected in E7 than in E6 of HPV 16, which in agreement with a previous report [21]. In addition to the numerical difference, only two variants identified—A712 (H51N) and G647 (N29S)—altered the E7 amino acid sequence and were detected at a very low frequency in southern Mexico. However, all the E6 variants showed amino acid changes [21].

Mutations in E7 herein identified were mostly synonymous (G666A, T678C, T732C, C765T, T789C, and T795G). Although these mutations do not change the amino acid residues of the oncoprotein, they are present at a high frequency in CC as is the case with the E7-C732/C789/G795 variant, which is associated with a 3.79 times higher risk of developing this pathology than compared with the E7-prototype. Moreover, E7-C732/C789/G795 was the second most frequent variant after the E7-prototype. What is noteworthy is that these three silent mutations were previously reported in Panama, United States, Suriname, Thailand, and India where they were found in patients with CC [26,27,28,29,30]. However, these reports did not evaluate the potential associations between the C732/C789/G795 variants in E7 of HPV 16 and the development of CC. The E7 variant that was previously reported to be associated with CC was E7-G647 [24,31], while in this study it was E7-C732/C789/G795. Although synonymous mutations do not modify the amino acid sequence of the oncoprotein, they may affect the structure, splicing, or stability of the RNA [32,33], thus contributing to oncogenicity; however, further studies are required to evaluate the precise effects. Taken together, these data emphasize the importance of analyzing local E7 variants and their association with the development of CC.

The distribution of HPV 16 E7 variants differs worldwide. In the American continent, the most reported variant is E7-C732/C789/G795; in Africa, it is E7-C789/G795; in Asia, they are E7-G647/C846, E7-G647, E7-C646, E7-A666, and E7-C732/C789/G795; and in Europe they are E7-C732/T749/G795, E7-G822, and E7-C789/G795 (Table 4). In this study, the most frequent E7 variant was E7-C732/C789/G795, which concurs with what has been reported in other countries in the American continent.

The complementary and synergistic effects of the HPV 16 oncoproteins E6 and E7 play a critical role in malignant transformation and cell immortalization [1]. There is extensive evidence that HPV 16 E6 and E7 mutations influence persistence, progression to precursor lesions, and the development of CC [12]. The differential oncogenic risk among variants of each oncoprotein has been studied; however, it is also necessary to analyze the oncogenic risk of combinations of E6/E7 variants.

One of the few studies on the oncogenic risk of the E6/E7 variant combination in HPV 16 focused on Korean women, in which the E6 T178G and E7 A647G (As*E7-G647) combination was the most prevalent and showed a significant association with the severity of cervical neoplasia [24]. In the present study, we found that individuals carrying the AA-a*E7-C732/C789/G795 combination possess the highest risk of CC (OR = 110), further suggesting that there may be a synergistic effect between the loci, as is evidenced by the lower risk for each locus individually (OR = 69.01 for AA-a of E6 [21] and 3.79 for E7-C732/C789/G795 in this study). The combinations of AA-c*E7-C732/C789/G795 and E-A176/G350*E7-prototype were also related to the risk of progression of precursor lesions to CC.

In conclusion, this is the first study of E7 variants of HPV 16 in Mexican women. Notably, E7-C732/C789/G795, a variant with three silent mutations, was associated with the risk of developing CC. These results highlight the importance of analyzing combinations of E6 and E7 variants in HPV 16 and suggest that AA-a*E7-C732/C789/G795, AA-c*E7-C732/C789/G795, and E-A176/G350*E7-prototype can be useful markers for predicting the progression of premalignant lesions and CC development.

## 4. Materials and Methods

### 4.1. Clinical Specimens and Diagnosis

The study enrolled 190 women, aged between 17 and 97 years old, who had all been identified as HPV 16 positive and were all residents of the Guerrero state in southern Mexico. Overall, 61 samples were from patients with CC, 6 samples were from patients with HSIL, 68 samples were from patients with LSIL, and 55 samples were from patients with no intraepithelial lesions. Cervical samples were obtained between 1997 and 2015 from cervical biopsies with histological diagnosis according to the FIGO classification system [65] or from cervical scrapes, with cytological diagnosis according to the Bethesda system [66]. Cervical scrapes (LSIL and no intraepithelial lesions) were collected using cytobrushes in a lysis buffer (10 mM Tris pH 8.0, 20 mM EDTA pH 8.0 and 0.5% sodium dodecyl sulfate) and the biopsies of women with HSIL or CC were immediately eluted in phosphate-buffered saline and stored at −70 ℃ until DNA extraction. DNA extraction was performed as previously described [67]. Briefly, the SDS-proteinase K-phenol-chloroform method [68] was used for DNA extraction from cervical scrapes and biopsies and the DNA was stored at −20 ℃ until analysis. The DNA integrity was evaluated by the amplification of the beta-globin gene [69]. All women without HPV 16 infection or from whom amplification of E6 and E7 genes was not obtained were excluded from the analysis. This study was approved by the Bioethical Committee of the Autonomous University of Guerrero and informed consent was obtained from all participants when the samples were collected.

### 4.2. Identification of HPV 16 E6/E7 Bicistronic Variants

E6 variants were identified as previously described [21]. HPV 16 E7 gene was amplified using the primers E7-F509 (5′-TGTATGTCTTGTTGCAGA-3′) and E7-R917 (5′-CATCCATTACATCCCGTA-3′), which amplify a 409 bp region. Polymerase chain reaction (PCR) was performed in a final volume of 25 µL containing the DNA template, 2.5 µL of 10× PCR buffer, 0.3 mM dNTPs, 1 mM MgCl_2_, 0.36 µM of each primer, and 0.7 U Maxima Hot Start Taq DNA Polymerase (EP0602; Thermo Fisher Scientific, Waltham, MA, USA). For positive and negative controls, reactions with 300 ng of DNA from SiHa or CaSki cells and H_2_O, respectively, were performed. PCR was performed using a Mastercycler EP Gradient (Eppendorf, Hamburg, Germany) as follows: initial denaturation at 95 ℃ for 4 min, followed by 40 cycles of 60 s at 95 ℃ for denaturation, 30 s at 57 ℃ for E6 and 51 ℃ for E7 for annealing, and 30 s at 72 ℃ for extension, with a final extension at 72 ℃ for 5 min. Correct amplification products were verified using DNA electrophoresis in an agarose gel.

Sequencing was performed using Big Dye Terminator V3.1 Cycle Sequencing (Applied Biosystems, Waltham, MA, USA) and the PCR products were purified using 75% isopropanol and loaded onto an ABI Prism 310 Genetic Analyzer (Applied Biosystems) according to the manufacturer’s instructions. All sequences were analyzed and aligned using Finch TV v. 1.4.0 (Geospiza Inc., Denver, CO, USA) and William Pearson’s LALIGN (https://embnet.vital-it.ch/software/LALIGN_form.html, accessed on 19 June 2021), respectively. The FASTA sequence of the reference HPV 16 NC 1526.3 was used for the alignment and for the identification of E7 variants. Sequence data are publicly available in the GenBank database (accession numbers: MW452666-MW452855).

### 4.3. Statistical Analysis

Data was analyzed using the STATA v.11 software (StataCorp, College Station, TX, USA). Chi-square test was performed for analyzing E7 and E6/E7 genetic variants and lesions. The age-adjusted ORs and 95% CI were calculated for the E7 variant and E6/E7 bicistronic variants. Statistical significance was set at *p* < 0.05.

## Figures and Tables

**Table 1 pathogens-10-00773-t001:** Clinical and demographic characteristics of patients included in the study.

	Non IL		LSIL		HSIL		CC		Total		*p*
	n	%	n	%	n	%	n	%	n	%	
**Mean age**	35.9		30.2		36.3		50.5		38.6		<0.005
**Age**											<0.005
<31	25	45.45	40	58.82	3	50.00	3	4.92	71	37.37	
31–40	12	21.82	19	27.94	1	16.67	13	21.31	45	23.68	
41–51	9	16.36	7	10.29	2	33.33	19	31.15	37	19.47	
>51	9	16.36	2	2.94	0	0	26	42.62	37	19.47	
**Age at menarche**											0.363
<14	32	58.18	43	63.23	3	50	31	50.82	109	57.37	
14–16	23	41.82	21	30.88	3	50	23	37.70	70	36.84	
>16	0	0	3	4.41	0	0	4	6.56	7	3.68	
Did not answer	0	0	1	1.47	0	0	3	4.92	4	2.11	
**Age at menopause**											<0.005
<45	3	5.45	1	1.47	1	16.67	11	18.03	16	8.42	
45–49	7	12.73	2	2.94	0	0	13	21.31	22	11.58	
>49	4	7.27	0	0	1	16.67	4	6.56	9	4.74	
No menopause	39	70.91	63	92.65	4	66.67	20	32.79	126	66.32	
Did not answer	2	3.64	2	2.94	0	0	13	21.31	17	8.95	
**Age at first sexual intercourse**											<0.005
<16	5	9.09	6	8.82	0	0	18	29.51	29	15.26	
16–20	36	65.45	37	54.41	5	83.33	32	52.46	110	57.89	
>20	14	24.45	24	35.29	1	16.67	8	13.11	47	24.74	
Did not answer	0	0	1	1.47	0	0	3	4.92	4	2.11	
**Geographical region of Guerrero**											<0.005
Costa Chica	0	0	1	1.47	0	0	7	11.48	8	4.21	
Costa Grande	0	0	0	0	0	0	9	14.75	9	4.74	
La Montaña	3	5.45	6	8.82	0	0	5	8.20	14	7.37	
Tierra Caliente	1	1.82	0	0	0	0	2	3.28	3	1.58	
Acapulco	5	9.09	6	8.82	0	0	11	18.03	22	11.58	
Zona Centro	41	74.55	51	75.00	6	100	10	16.39	108	56.84	
Zona Norte	2	3.64	3	4.41	0	0	9	14.75	14	7.37	
Did not answer	3	5.45	1	1.47	0	0	8	13.11	12	6.32	
Total	55	100	68	100	6	100	61	100	190	100	

Abbreviations: Non IL, non intraepithelial lesions; LSIL, low-grade squamous intraepithelial lesions; HSIL, high-grade squamous intraepithelial lesions; CC, cervical carcinoma.

**Table 2 pathogens-10-00773-t002:** Frequency of HPV 16 E7 genetic variants and their association with cervical carcinoma and precursor lesions in women from southern Mexico.

	E7 Region									Frequency															
	647	666	678	712	732	765	789	795		Non IL		LSIL				HSIL				CC				Total	
Variants	A	G	T	C	T	C	T	T	Aa Change in E7 ^a^	n	%	n	%	OR (95% CI)	*p*	n	%	OR (95% CI)	*p*	n	%	OR (95% CI)	*p*	n	%
E7-prototype	-	-	-	-	-	-	-	-	-	48	87.27	53	77.94	1 ^b^	-	3	50.0	1 ^b^	-	38	62.29	1 ^b^	-	142	74.74
E7-C732/C789/G795	-	-	-	-	c	-	c	g	-	7	12.72	8	11.76	1.03 (0.35–3.07)	0.950	3	50.0	6.86 (1.15–40.9)	0.35	21	34.42	3.79 (1.46–9.85)	0.006	39	20.53
E7-A666	-	a	-	-	-	-	-	-	-	0	0.00	2	2.94	-	-	0	0.00	-	-	2	3.28	-	-	4	2.11
E7-A712	-	-	-	A	-	-	-	-	H51N	0	0.00	2	2.94	-	-	0	0.00	-	-	0	0.00	-	-	2	1.05
E7-C678	-	-	c	-	-	-	-	-	-	0	0.00	1	1.47	-	-	0	0.00	-	-	0	0.00	-	-	1	0.53
E7-T765	-	-	-	-	-	t	-	-	-	0	0.00	1	1.47	-	-	0	0.00	-	-	0	0.00	-	-	1	0.53
E7-G647/C789/G795	G	-	-	-	-	-	c	g	N29S	0	0.00	1	1.47	-	-	0	0.00	-	-	0	0.00	-	-	1	0.53
Total	1	4	1	2	39	1	40	40		55	100	68	100			6	100			61	100			190	100

Note: Capital letters indicate the mutations that generate amino acid change and lowercase letters indicate mutation without amino acid change. Abbreviations: Non IL, non intraepithelial lesions; LSIL, low-grade squamous intraepithelial lesions; HSIL, high-grade squamous intraepithelial lesions; CC, cervical carcinoma; CI, confidence interval; OR, odd ratio. ^a^, the amino acid change prediction was made on HPV 16 reference, sequence NC 1526.3. ^b^, indicates reference category (E7-Prototype).

**Table 3 pathogens-10-00773-t003:** Frequency of combinations of E6 and E7 variants in the HPV 16 E6/E7 bicistron and its associations with cervical carcinoma and precursor lesions.

			Non IL		LSIL				HSIL				CC				Total	
Sublineage	E6 Variants	E7 Variants	n	%	n	%	OR (95% IC)	*p*	n	%	OR (95% IC)	*p*	n	%	OR (95% IC)	*p*	n	%
**A1**			13	23.6	7	10.3			1	16.7			3	4.9			24	12.6
	E-prototype	E7-prototype	10	18.2	7	10.3	**1 ^a^**	-	0	0	-	-	1	1.6	1 ^a^	-	18	9.5
	E-prototype	E7-732/C789/G795	1	1.8	0	0	-	-	1	16.7	-	-	0	0	-	-	2	1.1
	E-G131	E7-prototype	1	1.8	0	0	-	-	0	0	-	-	2	3.3	20 (0.85–471.57)	0.063	3	1.6
	E-G131	E7-732/C789/G795	1	1.8	0	0	-	-	0	0	-	-	0	0	-	-	1	0.5
**A2**			38	69.1	51	75			2	33.3			36	59			127	66.8
	E-G350	E7-prototype	23	41.8	29	42.6	1.8 (0.59–5.47)	0.29	2	33.3	-	-	16	26.2	6.95 (0.8–59.9)	0.08	70	36.8
	E-G350	E7-A666	0	0	2	2.9	-	-	0	0	-	-	2	3.3	-	-	4	2.1
	E-G350	E7-A712	0	0	1	1.5	-	-	0	0	-	-	0	0	-	-	1	0.5
	E-G350	E7-T765	0	0	1	1.5	-	-	0	0	-	-	0	0	-	-	1	0.5
	E-G350	E7-732/C789/G795	2	3.6	0	0	-	-	0	0	-	-	1	1.6	5 (0.21–117.89)	0.32	3	1.6
	E-C109/G350	E7-prototype	0	0	1	1.5	-	-	0	0	-	-	2	3.3	-	-	3	1.6
	E-G110/G350	E7-prototype	1	1.8	1	1.5	-	-	0	0	-	-	0	0	-	-	2	1.1
	E-G131/G350	E7-prototype	0	0	0	0	-	-	0	0	-	-	1	1.6	-	-	1	0.5
	E-131/C188/G350	E7-prototype	1	1.8	1	1.5	-	-	0	0	-	-	0	0	-	-	2	1.1
	E-A176/G350	E7-prototype	3	5.5	2	2.9	0.95 (0.12–7.28)	0.96	0	0	-	-	7	11.5	23.33 (1.99–273.3)	0.012	12	6.3
	E-A176/G350	E7-732/C789/G795	0	0	0	0	-	-	0	0	-	-	1	1.6	-	-	1	0.5
	E-C182/G350	E7-prototype	1	1.8	1	1.5	-	-	0	0	-	-	0	0	-	-	2	1.1
	E-T182/G350	E7-prototype	0	0	0	0	-	-	0	0	-	-	1	1.6	-	-	1	0.5
	E-C183/G350	E7-prototype	3	5.5	1	1.5	-	-	0	0	-	-	0	0	-	-	4	2.1
	E-G185/G350	E7-prototype	0	0	0	0	-	-	0	0	-	-	1	1.6	-	-	1	0.5
	E-A188/G350	E7-prototype	0	0	0	0	-	-	0	0	-	-	1	1.6	-	-	1	0.5
	E-C188/G350	E7-prototype	2	3.6	5	7.4	-	-	0	0	-	-	3	4.9	-	-	10	5.3
	E-C188/G350	E7-C678	0	0	1	1.5	-	-	0	0	-	-	0	0	-	-	1	0.5
	E-C188/G350	E7-A712	0	0	1	1.5	-	-	0	0	-	-	0	0	-	-	1	0.5
	E-C188/G310/G350	E7-prototype	1	1.8	1	1.5	-	-	0	0	-	-	0	0	-	-	2	1.1
	E-G189/T256/G350	E7-prototype	0	0	1	1.5	-	-	0	0	-	-	0	0	-	-	1	0.5
	E-G257/G350	E7-prototype	0	0	1	1.5	-	-	0	0	-	-	0	0	-	-	1	0.5
	E-C442/G350	E7-prototype	1	1.8	0	0	-	-	0	0	-	-	0	0	-	-	1	0.5
	E-G535/G350	E7-prototype	0	0	1	1.5	-	-	0	0	-	-	0	0	-	-	1	0.5
**C1**			0	0	1	1.5			0	0			0	0			1	0.5
	Af2-a/C109/G403	E7-647/C789/G795	0	0	1	1.5	-	-	0	0	-	-	0	0	-	-	1	0.5
**D2**			4	7.3	9	13.2			3	50			22	36.1			38	20.0
	AA-a	E7-prototype	1	1.8	1	1.5	-	-	0	0	-	-	0	0	-	-	2	1.1
	AA-a	E7-732/C789/G795	1	1.8	5	7.4	7.14 (0.68–75.22)	0.10	2	33.3	-	-	11	18	110 (6.04–2001.3)	0.001	19	10.0
	AA-c	E7-prototype	0	0	0	0	-	-	1	16.7	-	-	3	4.9	-	-	4	2.1
	AA-c	E7-732/C789/G795	2	3.6	3	4.4	2.14 (0.28–16.37)	0.46	0	0	-	-	7	11.5	35 (2.63–465.37)	0.007	12	6.3
	AA-c/G185	E7-732/C789/G795	0	0	0	0	-	-	0	0	-	-	1	1.6	-	-	1	0.5
	Total		55	100	68	100			6	100			61	100			190	100

Abbreviations: Non IL, non intraepithelial lesions; LSIL, low-grade squamous intraepithelial lesions; HSIL, high-grade squamous intraepithelial lesions; CC, cervical carcinoma; CI, confidence interval; OR, odd ratio. ^a^ Indicates reference category (E-prototype*E7-Prototype).

**Table 4 pathogens-10-00773-t004:** Most frequent HPV 16 E7 variants reported around the world.

				E7 HPV16 Variant				E7-prototype					
Variant	Total	CC	Non-IL/SIL	Total		CC		Total		CC		Country	Reference
	N	N	N	N	%	N	%	N	%	N	%		
**Africa**													
E7-C789/G795	21	21	0	9	43	9	43	3	14	3	14	Tanzania	Eschle et al., 1992 [34]
E7-C789/G795	5	3	2	5	100	3	100	0	0	0	0	Uganda	Buonaguro et al., 2000 [35]
E7-C789/G795	13	13	0	13	100	13	100	0	0	0	0	Congo	Boumba et al., 2015 [36]
**America**													
E7-C732/C789/G795	190	61	129	39	21	21	34	142	75	38	62	Mexico	This study
E7-C732/C789/G795	100	0	100	16	16	0	0	72	72	0	0	EU(Texas)	Swan et al., 2005 [28]
E7-C732/C789/G795	25	25	0	3	12	3	12	15	60	15	60	Suriname	De Boer et al., 2004 [27]
E7-C732/C789/G795	15	ND	ND	3	20	ND	ND	8	53	ND	ND	Panama, EU	Icenogle et al., 1991 [26]
E7-A712	12	0	12	2	17	0	0	6	50	0	0	Uruguay	Ramas et al., 2018 [37]
E7-C789/G795	11	11	0	6	55	6	55	3	27	3	27	Barbados	Smits et al., 1994 [38]
E7-C732	52	ND	ND	7	13	ND	ND	ND	ND	ND	ND	Costa Rica	Safaenian et al., 2010 [39]
E7-C789	52	ND	ND	8	15	ND	ND	ND	ND	ND	ND	Costa Rica	Safaenian et al., 2010 [39]
E7-G795	52	ND	ND	8	15	ND	ND	ND	ND	ND	ND	Costa Rica	Safaenian et al., 2010 [39]
**Asia**													
E7-G647/C846	292	0	292	125	43	0	0	125	43	0	0	China	Zhang et al., 2015 [40]
E7-G647/C846	167	53	114	126	75	37	70	7	4	2	4	China	Ding et al., 2010 [41]
E7-G647/C846	98	0	98	63	64	0	0	14	14	0	0	China	Zhao et al., 2020 [42]
E7-G647/C846	97	0	97	62	64	0	0	13	13	0	0	China	Cao et al., 2016 [43]
E7-G647/C846	76	76	0	26	34	26	34	15	20	15	20	China	Sun et al., 2012 [44]
E7-G647/C846	47	47	0	29	62	29	62	0	0	0	0	China	Wu et al., 2006 [23]
E7-G647/C846	58	38	20	36	62	26	68	ND	ND	1	3	China	Yang et al., 2014 [45]
E7-G647/C846	157	43	114	128	82	41	95	4	3	ND	ND	Taiwan	Chang et al., 2013 [31]
E7-G647/C846	15	9	6	6	40	5	56	3	20	2	22	Japan	Fujinaga et al., 1994 [46]
E7-G647/C846	31	0	31	18	58	0	0	2	10	0	0	Japan	Ishizaki et al., 2013 [47]
E7-G647/C846	24	0	24	5	21	0	0	18	75	0	0	Philippines	Ishizaki et al., 2013 [47]
E7-G647/C846	24	0	24	20	83	0	0	0	0	0	0	Vietnam	Ishizaki et al., 2013 [47]
E7-G647/C846	9	9	0	3	33	3	33	3	33	3	33	Thailand	Chansaenroj et al., 2012 [29]
E7-C732/C789/G795	9	9	0	3	33	3	33	3	33	3	33	Thailand	Chansaenroj et al., 2012 [29]
E7-G647	141	25	116	63	45	18	72	55	39	5	20	Korea	Lee et al., 2011 [24]
E7-G647	133	31	102	118	89	55	41	15	11	2	7	Korea	Park et al., 2016 [48]
E7-G647	56	0	56	36	64	0	0	7	13	0	0	Korea	Choi et al., 2007 [49]
E7-G647	42	27	15	25	60	19	70	8	19	3	11	Korea	Song et al., 1997 [50]
E7-G647	143	81	62	54	38	29	36	ND	ND	ND	ND	India	Radhakrishna Pillai et al., 2002 [51]
E7-G647	43	43	0	1	2	1	2	42	98	42	98	China	He et al., 2016 [52]
E7-G647	31	0	31	24	77	0	0	ND	ND	0	0	China	Zhao et al., 2019 [53]
E7-G647	31	31	0	19	61	19	61	9	29	9	29	Thailand	Vaeteewoottacharn et al., 2003 [54]
E7-C646	100	0	100	21	21	0	0	ND	ND	0	0	China	Zhang et al., 2016 [55]
E7-A666	52	52	0	20	39	20	39	8	15	8	15	China	Shang et al., 2011 [56]
E7-A666	22	22	0	15	68	15	68	2	9	2	9	Indonesia	De Boer et al., 2004 [27]
E7-C789/G795	60	60	0	3	5	3	5	52	87	52	87	India (North)	Pande et al., 2008 [57]
E7-C732/C789/G795	110	94	16	12	11	11	12	95	86	80	85	India	Mazumder Indra et al., 2011 [30]
**Europe**													
E7-C732/T749/G795	124	28	96	8	7	4	14	103	83	19	68	Romania	Plesa et al., 2014 [58]
E7-G822	112	37	75	15	13	4	11	89	80	32	87	Germany	Nindl et al., 1999 [59]
E7-G822	61	31	30	4	7	2	7	55	90	27	87	Sweden	Hu et al., 2001 [60]
E7-G822	40	40	0	2	5	2	5	35	88	35	88	Slovenia	Vrtačnik Bokal et al., 2010 [61]
E7-C789/G795	90	36	54	10	11	4	11	71	79	24	67	Italy	Tornesello et al., 2004 [62]
E7-C789/G795	14	0	14	1	7	0	0	13	93	0	0	Italy	Garbuglia et al., 2007 [63]
E7-C789/G795	27	27	0	3	11	3	11	21	78	21	78	Netherlands	De Boer et al., 2004 [27]
E7-C732/C789/G795	29	0	29	1	3	0	0	27	93	0	0	Italy	Cento et al., 2009 [64]
E7-T616	8	3	5	1	13	1	33	7	88	2	67	Netherlands	Smits et al., 1994 [38]
-	6	6	0	0	0	0	0	6	100	6	100	Germany	Eschle et al., 1992 [34]

## Data Availability

Data generated and analyzed in this study are available upon request from the authors. The sequence data are available in GenBank (https://www.ncbi.nlm.nih.gov/genbank/, accessed on 19 June 2021) with the accession numbers MW452666-MW452855.

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
