# Peer review of "E6/E7 Variants of Human Papillomavirus 16 Associated with Cervical Carcinoma in Women in Southern Mexico"

_pathogens, 2021, doi:10.3390/pathogens10060773_

Round 1
Reviewer 1 Report
I have no major issues with the manuscript, only a few minor English corrections to suggest.
Line 21: Replace “its” with “CC”
Line 22: Delete comma after “Mexico”
Line 25: Add the abbreviation for “high-grade squamous intraepithelial lesions” (HSIL) here or in the first mention of the term in the “Results” section, on line 87
Line(s) 25/26: Add the abbreviation for “low-grade squamous intraepithelial lesions” (LSIL) here or in the first mention of the term in the “Results” section, on line 88
Line(s) 26/29: Rework the sentence: “Carriers of the E7 variant E7- 26 C732/C789/G795, which has three silent mutations, had the highest risk of CC (OR = 3.79, 95% CI = 27 1.46–9.85). AA-a*E7-C732/C789/G795, an E6/E7 bicistron, conferred the greatest increased risk of CC 28 (OR = 110, 95% CI = 6.04–2001.3), followed by AA-c*E7-C732/C789/G795 and A176/G350*E7-p.” to make it clear that the C732/C789/G795 variant has the highest risk of CC for the E7 variants discovered and the AA-a*E7-C732/C789/G795, the E6/E7 bicistron mutant, has the highest overall risk of CC for the variants discovered.
Line 32: Delete the second period
Line 37: Add “main” before “cause”
Line 40: Add a colon before “promote”
Table 1: Is the discontinuous lines under “LSIL”, “CC”, “p” deliberate? Also, the P values in the last column look off-set
Table 2: There are more discontinuous lines under the E7 variants and before the total values in the left-hand panel. Also, “Non IL”, and “HSIL”, and “Total” headers for the right-hand panel also have discontinuous lines – are these deliberate?
Table 3: “Non IL”, and “HSIL”, and “Total” headers are have discontinuous lines underneath them, if that is deliberate, that is fine, just want to make sure this is a conscious choice by the authors.
Author Response
Dear reviewer 1
Thank you for your letter dated June 13th, 2021, concerning our manuscript, entitled " E6/E7 variants of human papillomavirus 16 associated with cervical carcinoma in women in southern Mexico” by Antaño-Arias R et al. (manuscript number: Pathogens-1255368).
After reviewing the content of your letter and the suggestions provided by the reviewers, we now submit a revised version of our manuscript, to which we have made several changes according to your suggestions, this has undoubtedly greatly improved our manuscript and attempt to clarify, as much as possible, each point raised.

Reviewer 2 Report
This manuscript is well-written and is a valuable paper that showing that AA-a*E7-C732/C789/G795, AA-c*E7-C732/C789/G795, and A176/G350*E7-p may be useful markers for predicting the development of cervical cancer. This is likely to be interesting study, although there are several points to be revised in the current form.
- Cervical samples are noted to have used a cervical biopsy or cervical scrape. Is the sample analyzed different for each patient? Is the sample analyzed different for each patient? Does using different samples affect the analysis of E7 variants? FFPE tissue fragments from cervical biopsy are affected by DNA fragmentation. Please comment on the reliability of the analysis of the E7 variant by FFPE. Also, was the tissue sample of this study microdissected only intraepithelial lesion cells, or was it the whole tissue? Please explain the method of microdissection and DNA extraction.
- Most LSIL cases were young women. These cases are assumed to be multiple HPV infections. I found it difficult to assess the risk of cervical cancer in multiple infected samples. Were all the samples used in this study a single infection? Please describe the HPV infection status (single or multiple infection) of cervical samples.
- The authors also detected variants of E7-C732 / C789 / G795 in non IL and LSIL collected between 1995 and 2015. Did these cases develop to cervical cancer? Please show any cases for which follow-up results are known.
- 95% IC in the 10th row from the bottom of Table 3 was missing.
Author Response
Dear reviewer 2
Thank you for your letter dated June 13th, 2021, concerning our manuscript, entitled " E6/E7 variants of human papillomavirus 16 associated with cervical carcinoma in women in southern Mexico” by Antaño-Arias R et al. (manuscript number: Pathogens-1255368).
After reviewing the content of your letter and the suggestions provided by the reviewers, we now submit a revised version of our manuscript, to which we have made several changes according to your suggestions, this has undoubtedly greatly improved our manuscript and attempt to clarify, as much as possible, each point raised.

Reviewer 3 Report
The manuscript entitled: “E6/E7 variants of human papillomavirus 16 associated with cervical carcinoma in women in southern Mexico” is a descriptive work where the genomic variations of HPV16 E7 genes in 190 cervical samples (scrapes and cervical biopsias) of a population in southern Mexico was analyzed, including normal epithelia, low- and high-grade squamous intraepithelial lesions and cervical cancer. The same group of samples was analyzed for E6 variations in a previous publication from the same group.
Sequencing of E7 PCR amplified fragments was performed and clear description of mutations found for E7 gene in the studied population is shown.
The risk for premalignant lesions or cervical cancer was evaluated related to E7 variations combined with the already published E6 variations. One of the found E7 variant had the highest risk for cervical cancer (E7-C732/C789/G795), although the respective mutations did not drive to changes in aminoacids. This was also the most frequent E7 variant found in the studied population.
In general the manuscript is well written. The discussion is well conducted where the different HPV16 E7 variants found in other populations are described.
The authors conclude that the analysis of variations in E6/E7 bicistron could be a useful marker for risk of progression to premalignant cervical lesions and cervical cancer development.
This manuscript has a relevant epidemiological contribution, related to the intratype variations of HPV16 in a specific population.
Author Response
Dear reviewer 3
Thank you for your letter dated June 13th, 2021, concerning our manuscript, entitled " E6/E7 variants of human papillomavirus 16 associated with cervical carcinoma in women in southern Mexico” by Antaño-Arias R et al. (manuscript number: Pathogens-1255368).
After reviewing the content of your letter, we have read your comments and appreciate the time you took to review and evaluate every aspect of our work.